# Vibration Control of Diamond Nanothreads by Lattice Defect Introduction for Application in Nanomechanical Sensors

**DOI:** 10.3390/nano11092241

**Published:** 2021-08-30

**Authors:** Xiao-Wen Lei, Kazuki Bando, Jin-Xing Shi

**Affiliations:** 1Department of Mechanical Engineering, University of Fukui, 3-9-1 Bunkyo, Fukui 910-8507, Japan; paille1003@gmail.com; 2Department of Production Systems Engineering and Sciences, Komatsu University, Nu 1-3 Shicyomachi, Komatsu 923-8511, Japan; jinxing.shi@komatsu-u.ac.jp

**Keywords:** diamond nanothreads, lattice defects, molecular dynamics, continuum mechanics

## Abstract

Carbon nanomaterials, such as carbon nanotubes (CNTs) and graphene sheets (GSs), have been adopted as resonators in vibration-based nanomechanical sensors because of their extremely high stiffness and small size. Diamond nanothreads (DNTs) are a new class of one-dimensional carbon nanomaterials with extraordinary physical and chemical properties. Their structures are similar to that of diamond in that they possess ${\mathrm{sp}}^{3}$-bonds formed by a covalent interaction between multiple benzene molecules. In this study, we focus on investigating the mechanical properties and vibration behaviors of DNTs with and without lattice defects and examine the influence of density and configuration of lattice defects on the two them in detail, using the molecular dynamics method and a continuum mechanics approach. We find that Young’s modulus and the natural frequency can be controlled by alternating the density of the lattice defects. Furthermore, we investigate and explore the use of DNTs as resonators in nanosensors. It is shown that applying an additional extremely small mass or strain to all types of DNTs significantly changes their resonance frequencies. The results show that, similar to CNTs and GSs, DNTs have potential application as resonators in nano-mass and nano-strain sensors. In particular, the vibration behaviors of DNT resonators can be controlled by alternating the density of the lattice defects to achieve the best sensitivities.

## 1. Introduction

Following the rapid development of nanotechnology over the past several decades, carbon nanomaterials, e.g., one-dimensional carbon nanotubes (CNTs), two-dimensional graphene sheets (GSs), and one-dimensional carbyne, have been applied or proposed as resonators in nanomechanical sensors because of their excellent mechanical, optical, and electrical properties [1,2,3,4]. Poncharal et al. [1] developed a nano-mass sensor, using a CNT resonator, which can measure extremely small masses in the picogram to femtogram range. Deflections of a cantilevered CNT resonator were electrically induced by transmission electron microscopy, and the resonated frequency was determined, using the deflected contours. Bunch et al. [2] fabricated nanoelectromechanical systems using GSs as resonators, whose fundamental resonant frequency vibrations were electrically or optically actuated and optically detected by interferometry. The fabricated GS resonators were considered to be ideally suited for application in nano-mass, nano-force, and charge sensors. A nano-mass sensor using carbyne as the resonator was proposed by Shi et al. [3] for measuring tiny weights by theoretically determining the resonant frequency shifts. More details and discussions of carbon nanomaterials–based nanomechanical sensors, particularly nano-mass and nano-force sensors, can be found in a recent review [4]. Based on this review, reducing the dimensions and increasing the stiffness of carbon nanomaterials–based resonators enhances the sensitivities of the corresponding nanomechanical sensors. The present study aims to investigate new frequency-based nano-mass and nano-strain sensors, using diamond nanothreads (DNTs) as resonators.

DNTs, also named carbon nanothreads, are a new class of carbon nanomaterials with a one-dimensional ${\mathrm{sp}}^{3}$-bonded structure, which were first synthesized by Fitzgibbons et al. [5] from benzene under solid-state high-pressure action. Recent studies have shown that DNTs possess excellent mechanical properties on similar levels to those of CNTs. Moreover, they have ultrathin CNT analogues [6,7,8,9,10,11,12,13,14], which make them suitable as resonators in nanomechanical sensors [15]. Roman et al. [6] determined the stiffness, strength, extension, and bending rigidity of DNTs as 850 GPa, 26.4 nN, 14.9%, and $5.35\times 10^{-28}$ Nm${}^{2}$, respectively, based on molecular dynamics (MD) simulations. Feng et al. [9] studied tensile and bending behaviors of lowest-energy DNTs by full atomistic first principles–based MD simulations. Based on the results, they concluded that the tensile stress–strain responses and bending stiffness of all DNTs are distinct because of their different morphologies; therefore, the mechanical properties of DNTs can be controlled by specifying their morphologies. Silveira and Muniz [10] investigated the mechanical properties of DNTs by performing first-principles calculation and determined that DNTs present strength and stiffness of 15.7 nN and 168 nN, respectively, which are similar to those of CNTs. They also conducted MD simulations for comparison and demonstrated that adopting the adaptive intermolecular reactive empirical bond-order (AIREBO) potential in MD simulations could predict accurately predict the mechanical properties of DNTs. Considering their ultrahigh mechanical properties, Duan et al. [15] proposed DNTs as resonators in nanomechanical sensors. They performed MD simulations to investigate their dynamic characteristics, and the results showed that DNT-based nano-mass sensors have an excellent mass resolution of $0.58\times 10^{-24}$ g, which is higher than those of CNTs- or GSs-based nano-mass sensors. Recently, there were some nanothreads that were produced by different methods, such as one-dimensional carbon nanothreads through modest-pressure polymerization of Furan [16], orientational order in nanothreads derived from thiophene [17], one-dimensional diamondoid polyaniline-like nanothreads from compressed crystal aniline [18], double core chromophore-functionalized nanothreads by compressing azobenzene [19], and carbon nitride nanothreads [20,21].

Nanomechanical sensors play an important role in the development of nanotechnology [22,23,24,25]. Particularly, frequency-based nano-mechanical sensors (e.g., nano-mass and nano-force/strain sensors) using CNTs or GSs as resonators have been investigated in numerous studies [26,27,28,29,30,31]. The mechanism of frequency-based nano-mass and nano-force/strain sensors is the determination of the resonant frequency shift of the nanoresonators under the action of unknown masses and forces. Hence, the mechanical properties of nanoresonators (such as stiffness and densities) are important to the sensitivities of the sensors. It was concluded that CNT-based nano-mass and nano-force sensors achieve a mass resolution of $10^{-21}$ g and a force detection of 2.5 nN, and GSs-based nano-mass sensors present a mass resolution of $10^{-24}$–$10^{-22}$ g, at least [4]. As mentioned above, because of the differences in the mechanical properties and dimensions of resonators, sensors present varied performance. Hence, successfully controlling the mechanical properties of resonators can lead to the realization of target sensitivities in nanomechanical sensors. It is well known that introducing lattice defects in carbon nanomaterials (e.g., CNTs [32,33,34], GSs [35,36,37,38], and DNTs [8,9,10,15,39]) can change or control their mechanical or electrical behaviors. Zhang et al. [32] simulated the fracture of CNTs containing one- and two-atom vacancies, using molecular mechanics calculations, which showed that the fracture strength of the defected CNTs was reduced by 20–30%. Shi et al. [38] performed the optimal shape design of GSs by introducing lattice defects to enhance their vibration behaviors. Wu et al. [40] reported that the electronic properties of DNTs can be adjusted by varying the density of lattice defects. Introducing lattice defects in carbon nanomaterials can control their mechanical behavior well. Consequently, DNTs exhibit outstanding properties as do other carbon nanomaterials, such as CNTs and GSs. For example, DNTs show a brittle to ductile transition characteristic by controlling the Stone-Wales (SW) defects [8]; for special structure of DNTs, the uniaxial stress can form single-crystalline packings of polymers, threads, and higher dimensional carbon networks [39].

Carbon nanomaterials with perfect lattice structures typically undergo local recombination to form lattice defects under high temperature. SW defects are frequently observed in low-dimensional carbon nanomaterials. In CNTs and GSs, under certain conditions, the single carbon–carbon bonds in their six-membered rings rotate $90^{\circ }$, transforming four six-membered rings into two five- and seven-membered rings each, which is called the SW defect. In DNTs, this occurs differently from CNTs and GSs. In a perfect DNT, two carbon–carbon bonds parallel to each other, shown in red in Figure 1a, rotate about the right carbon atom by $90^{\circ }$ in a clockwise direction. A SW defect is formed as two pairs of five-membered rings, shown in green in Figure 1b, are bonded with the rotated carbon–carbon bonds in a parallel relationship, shown in red in Figure 1b. The other carbon atoms are represented in gray, and the hydrogen atoms are shown in blue. The above structures of the perfect DNT and the DNT with a SW defect have the same number of carbon and hydrogen atoms. Owing to the SW defect, the straight DNT structure changes to an eccentric structure with periodicity. Because the perfect DNT structure is almost nonexistent in reality, the SW defect, as one of the most common lattice defects, can influence the physical and chemical characteristics of DNTs. In this study, we investigate the effects of the number and arrangement of SW defects on the mechanical properties of DNTs.

The nonlocal Timoshenko beam model has been used in many studies since Peddieson [41] applied Eringenś theory of nonlocal elasticity [42] in nanotechnology [43,44,45], bending [46,47], buckling [48], and vibrations of elastic nano-beams [49,50,51]. Moreover, nonlocal elasticity theory has been also adopted in the bending of beam elements in microelectromechanical and nanoelectromechanical system devices [52,53,54,55], such as carbon nanomaterials as mass sensors [53,54,55].

In this study, we aim to develop nanomechanical (mainly nano-mass and nano-strain type) sensors using DNTs as resonators and control their vibration behaviors by introducing lattice defects to realize the best sensitivities. We investigate the mechanical and vibration properties of perfect DNTs and DNTs with SW defects by MD simulation and the nonlocal Timoshenko beam theory to develop new nanomechanical sensors. In the remainder of this paper, subsequently, in Section 2, we introduce the adopted analytical methods, i.e., MD simulations and the nonlocal Timoshenko beam theory, for vibration analysis of DNT resonators. The tensile tests for determining the material properties of the DNTs as well as the vibration analysis of the DNT-based resonators for application in nano-mass and nano-strain sensors are discussed in Section 3. Moreover, the results of present work are confronted and compared with previous results. Finally, in Section 4, remarkable conclusions are drawn.

## 2. Methods

### 2.1. Molecular Dynamics Simulation

Figure 2 shows the MD simulation models of a perfect DNT and DNTs with three types of SW defects. The existence of SW defects induces a local eccentric structure in the DNTs and varies the projection length along the *x*-axis direction. To maintain complete lattices in both ends of the DNTs, the length of each simulation model in the *x*-axis direction is approximately 110.0 Å. The DNTs with *n*-isolated, *n*-double, and *n*-triple SW defects are represented as DNT-*n*, DNT-*n*d, and DNT-*n*t, respectively. We also analyze Polymer I [56], which has a structure in which only the SW defect part is expanded one-dimensionally. Enlarged views of the local structures are shown in the right box in Figure 2, and the color scheme is the same as in Figure 1.

In this study, we employ the classical MD method using LAMMPS [57] to analyze the mechanical and vibrational characteristics of a DNT with SW defects, whose interatomic interaction is expressed by the AIREBO potential [58]. The timestep is 1 fs. The cut-off distance is 1.95 Å [8]. To investigate only the mechanical effect in detail, the temperature condition is set as *T* = 5 K. Figure 3 shows the boundary conditions of the tensile and vibration analyses. The following are the analysis conditions: distance between two adjacent carbon atoms 1.52 Å and distance between carbon and hydrogen atoms 1.10 Å [5,6]. The differences in the analysis conditions of the tensile and vibration analyses are discussed below.

In the tensile analysis, the tensile speed is 0.01 Å/ps. The system temperature 5 K is stabilized considering an NPT ensemble with 100,000 steps.In the vibration analysis, an initial displacement is applied along the *z*-axis to the six-membered ring at the center in the *x*-axis direction, and subsequently the constraint is released to achieve free vibration.

In each analysis model, the central carbon atom in the *x*-axis direction is used as a reference, and a fast Fourier transform is applied on the displacement change with time in the *z*-axis coordinates to evaluate the primary mode. The procedure of the vibration analysis common for strain and mass application is presented below.

The relaxed structure of the analytical model is obtained, using the conjugate gradient method with accuracy $10^{-17}$ eV.The system temperature 5 K is stabilized, considering an NVT ensemble with 100,000 steps.The constraint is released after an initial displacement is applied to the six-membered ring centered in the *x*-axis direction, considering an NVT ensemble with 3,000,000 steps.Free vibration is performed in an NVE ensemble with 3,000,000 steps.

The following are differences in the procedure for the vibration analysis of the DNTs under an applied strain or mass.

In the vibration analysis of a DNT under an applied strain, the strain is applied immediately after the structure of the system stabilizes, after which an initial displacement is applied.In the vibration analysis of a DNT with an additional mass, the atoms at the center in the *x*-axis direction being set with additional mass are regarded as the equivalent mass atoms. The analysis is performed according to analysis steps with different equivalent mass atoms.

### 2.2. Nonlocal Timoshenko Beam Theory

Equations (Equation 1) and (Equation 2) show the governing equations of the continuum mechanics theory, using the nonlocal Timoshenko beam model.
(1)$$E_{A}I\frac{\partial^{2}\varphi }{\partial x^{2}}+kGA\left(\frac{\partial w}{\partial x}-\varphi \right)=\left[1-{\left(e_{0}a\right)}^{2}\frac{\partial^{2}}{\partial x^{2}}\right]\rho I\frac{\partial^{2}\varphi }{\partial t^{2}}$$
(2)$$kGA\left(\frac{\partial^{2}w}{\partial x^{2}}-\frac{\partial \varphi }{\partial x}\right)=\left[1-{\left(e_{0}a\right)}^{2}\frac{\partial^{2}}{\partial x^{2}}\right]\rho A\frac{\partial^{2}w}{\partial t^{2}}$$
where $E_{A}$ and *G* are Young’s modulus and the shear elastic modulus, *I* is the moment of inertia, *k* is the shear coefficient, *A* is the area of cross-section, $e_{0}a$ is the nonlocal coefficient, ρ denotes the density, and *x* and *t* indicate the longitudinal coordinate and the time, respectively. Moreover, the displacement $w\left(x,t\right)$ and the angle of rotation $\varphi \left(x,t\right)$ are given as follows.
(3)$$w\left(x,t\right)=\sin\frac{m\pi x}{L}\sin\omega t$$
(4)$$\varphi \left(x,t\right)=\cos\frac{m\pi x}{L}\sin\omega t$$

The following equations express the bridged (fixed-fixed) boundary conditions:(5)$$w\left(0,t\right)=w\left(L,t\right)=0$$
(6)$$\frac{\partial \varphi \left(0,t\right)}{\partial x}=\frac{\partial \varphi \left(L,t\right)}{\partial x}=0$$

For simplification of the calculation, the above can be expressed in matrix form as follows:(7)$$\left[\begin{array}{cc} L_{11} & L_{12}\\ L_{21} & L_{22} \end{array}\right]\left\{\begin{array}{c} \varphi \\ w \end{array}\right\}=0$$

Here, $L_{11}$–$L_{22}$ are as follows:(8)$$L_{11}=E_{A}I\frac{\partial^{2}}{\partial x^{2}}-kGA-\rho I\frac{\partial^{2}}{\partial t^{2}}+\rho I{\left(e_{0}a\right)}^{2}\frac{\partial^{4}}{\partial x^{2}\partial t^{2}}$$
(9)$$L_{12}=kGA\frac{\partial }{\partial x}$$
(10)$$L_{21}=-kGA\frac{\partial }{\partial x}$$
(11)$$L_{22}=kGA\frac{\partial^{2}}{\partial x^{2}}-\rho A\frac{\partial^{2}}{\partial t^{2}}+\rho A{\left(e_{0}a\right)}^{2}\frac{\partial^{4}}{\partial x^{2}\partial t^{2}}$$

Further calculation can also be expressed in matrix form as the following:(12)$$\left[\begin{array}{cc} L_{11} & L_{12}\\ L_{21} & L_{22} \end{array}\right]=0$$

Here, $L_{11}$–$L_{22}$ are given as follows:(13)$$L_{11}=-E_{A}I{\left(\frac{m\pi }{L}\right)}^{2}-kGA+\rho I\omega^{2}+\rho I{\left(e_{0}a\right)}^{2}{\left(\frac{m\pi }{L}\right)}^{2}\omega^{2}$$
(14)$$L_{12}=kGA\frac{m\pi }{L}$$
(15)$$L_{21}=kGA\frac{m\pi }{L}$$
(16)$$L_{22}=-kGA{\left(\frac{m\pi }{L}\right)}^{2}+\rho A\omega^{2}+\rho A{\left(e_{0}a\right)}^{2}{\left(\frac{m\pi }{L}\right)}^{2}\omega^{2}$$

Here, $E_{A}$ is Young’s modulus of a DNT obtained from the tensile analysis results, shear elastic modulus *G* is 267.2 GPa, shear coefficient *k* is 0.8, and nonlocal coefficient $e_{0}a$ is $4.65\times 10^{-7}$ Åfor matching the MD results. To calculate the first vibration mode of the DNTs, the half-wave frequency, *m*, is taken as 1. Because the DNTs are compressed by benzene rings containing six carbon atoms and six hydrogen atoms, the radius is 2.47 Å, cross-sectional area *A* is 19.15 Å${}^{2}$, volume is 38.81 Å${}^{3}$, density ρ is 0.0334 yg/Å${}^{3}$, moment of inertia *I* is 29.19 Å${}^{4}$, and length *L* in the *x*-axis direction of the analytical model is approximately 110.0 Å.

## 3. Results and Discussion

### 3.1. Determination of Material Properties

To obtain Young’s modulus of the DNTs with and without SW defects, we perform tensile tests based on MD simulation, and the stress–strain diagrams obtained from the analysis are shown in Figure 4 and Figure 5. From Figure 4, we can see that for the same strain, the stress of the DNT without SW defects is the highest, whereas that of Polymer I is the lowest with the largest fracture strain. The stress–strain curves of DNT-*n* lie almost between those of the DNT without SW defects and Polymer I, and DNT-*n* and DNT-*n*d (*n* = 2, 3, 4, 6, 9) are considered to have characteristics of both the perfect DNT and Polymer I. For both the DNT with isolated SW defects (see Figure 4) and DNT with double SW defects (see Figure 5), the breaking stress and breaking strain are significantly smaller than those of the perfect DNT and Polymer I. The results of DNT with SW defects are in agreement with the previous results in Ref. [8]. In the structures of the DNTs with SW defects, fracture occurs in the SW defect part, which suggests that stress concentration occurs in the SW defect local part in the DNTs with SW defects. We also consider the effect of the SW defect density on the mechanical behavior of DNTs. In the DNT with SW defects, there is a plastic region between the yield and breaking point, which enhances the breaking stress and breaking strain. In addition, we can find that as the SW defect density increases, the plastic region expands. The fracture strain increases monotonically with the increasing SW defect density, which has the same range and trend of Young’s modulus reported in Ref. [6]. The characteristics of Polymer I, which only consists of SW defects with relatively lower rigidity and higher ductility [11], are significantly different from those of the perfect DNT, owing to the differences in the configuration of the carbon atoms.

The relationship between Young’s modulus and the number of SW defects is shown in Figure 6; Young’s modulus decreases monotonically with the increasing SW defect density. The theoretical analysis by continuum mechanics in Figure 6 is derived in Equation (Equation 21).

We consider the CNT as a continuum beam and we use continuum mechanics to calculate Young’s modulus by Equation (Equation 21). The elongation of the whole structure λ in Equation (Equation 17) is expressed in Equation (Equation 18); the elongation of the DNT and the SW defects $\lambda_{\mathrm{DNT}}$ and $\lambda_{\mathrm{SW}}$ are given in Equations (17c) and (17d), respectively. Young’s modulus of Polymer I$E_{\mathrm{SW}}$ is 581.6 GPa, and Young’s modulus of the perfect DNT, $E_{\mathrm{DNT}}$, is 961.2 GPa. The length of DNT in the *x*-axis direction *L* is 110.0 Å, and the length of one of the SW defects in the *x*-axis direction $L_{\mathrm{SW}}$ is 6.3 Å. The correction value δ is −66.5 GPa, and *p* is the number of SW defects. From Figure 6, the largest difference between the analytical and the theoretical values is approximately 3.00% for DNT-4. Therefore, the stiffness can be estimated using the theoretical formula, and it is considered possible to control the stiffness and ductility of DNTs by tuning the SW defect density.
(17a)$$E=\frac{PL}{\lambda A}$$
(17b)$$\lambda =\lambda_{\mathrm{DNT}}+\lambda_{\mathrm{SW}}$$
(17c)$$\lambda_{\mathrm{DNT}}=\frac{P(L-pL_{\mathrm{SW}})}{E_{\mathrm{DNT}}A}$$
(17d)$$\lambda_{\mathrm{SW}}=\frac{P(pL_{\mathrm{SW}})}{E_{\mathrm{SW}}A}$$
(17e)$$E_{\left(p\right)}=\frac{E_{\mathrm{SW}}E_{\mathrm{DNT}}L}{\left(E_{\mathrm{DNT}}L_{\mathrm{SW}}-E_{\mathrm{SW}}L_{\mathrm{SW}}\right)p+E_{\mathrm{SW}}L}+\delta$$

### 3.2. Vibration Analysis

Figure 7 shows the natural frequencies obtained from the MD simulation and the nonlocal Timoshenko beam theory; clearly, the analytical and theoretical values present good agreement. The largest difference between the two values is found for DNT-3, and the relative error is approximately 3.91%. The results suggest that the natural frequencies of DNTs can be estimated using the simplified theoretical formula. In detail, the natural frequencies of all types of DNTs are within a high frequency range of 80–100 GPa, and they decrease monotonically as the SW defect density increases. Because in Figure 6, Young’s modulus also decreases monotonically with increasing the SW defect density, it is considered that the natural frequency of a DNT increases as its rigidity increases. Therefore, it is concluded that the frequencies of DNTs can be controlled by tuning the rigidity base on the SW defect density.

The relationships between the natural frequencies and strains applied on the DNTs with different types of SW defects are shown in Figure 8, Figure 9 and Figure 10. The structures of these DNTs with ε=0.00 are not straight, and the eccentricity is significantly affected by the SW defects, which can be intuitively seen in Figure 2. In addition, stress concentration occurs in the local parts of the SW defects, even in the relaxed structures. In all analytical models, the natural frequencies of the DNTs increase with increasing the applied strain. Moreover, when the DNTs are applied by the same strain (except ε=0.00), the frequencies of the DNTs with SW defects decrease monotonically with increasing the SW defect density. Figure 10 shows a comparison of the natural frequencies of the DNTs with an isolated SW defect and three continuous SW defects. DNT-6 and DNT-2*t* as well as DNT-9 and DNT-3*t* have the same numbers of SW defects, respectively, and their natural frequencies are almost similar. The results show that the influence of the number of SW defects is much greater than that of their configuration. Therefore, it is considered that the natural frequencies of DNTs with isolated SW defects and continuous SW defects can be controlled by the SW defect density because the SW defect density significantly affects the rigidity of DNTs. In addition, the frequency of a DNT can be adjusted by applying strain, and it is considered that it can be applied as a nanoscale strain sensor.

Figure 11 and Figure 12 show the relationship between the resonance frequencies and the added masses of the DNTs with an isolated SW defect and continuous SW defects. The approximate curve equation for the figures can be expressed as Equation (Equation 18) [59,60], which is applied for the first time in the DNT vibration analysis in this study.
(18)$$m=\alpha \left[{\left(\frac{f_{0}}{f_{m}}\right)}^{2}-1\right]$$
where *m* denotes the attached mass, $f_{0}$ is the natural frequency, $f_{m}$ is the resonant frequency with the attached mass, and α is the constant obtained by fitting the analysis results. For all analysis models, when the additional mass exceeds $10^{2}$ yg, the resonance frequency remarkably decreases as the additional mass increases. The resonant frequencies of all types of DNT-*n* present similar change trends; however, the amount of change in the resonance frequency for the same amount of mass change increases as the SW defect density decreases. Therefore, it can be inferred that a DNT with a low SW defect density can detect mass with high sensitivity. In addition, the analysis and approximate curved are in good agreement, and it is considered that the additional mass can be predicted using the simplified resonance frequency equation. Based on these results, we consider that DNTs have potential applications not only as nanoscale strain sensors, but also as nanoscale mass sensors.

## 4. Conclusions

In the present work, the mechanical properties and vibration behaviors of perfect DNTs and DNTs with isolated or continuous lattice defects were investigated, using the MD method and a continuum mechanics approach. We found that the results obtained by the two methods agreed with each other very well, suggesting that the simplified continuum mechanics equations could be used to estimate the behaviors of nanoscale material DNTs. In particular, the continuum mechanics could predict the mechanical properties of DNT simply. Moreover, the SW defect density had a major influence on the mechanical properties, and it is also suggested that the rigidity and ductility of DNTs could be controlled by the SW defect density. Furthermore, it was shown that the DNT frequencies also depend on the SW defect density and present one-to-one relationships with additional mass and applied strain. We consider that by controlling the SW defects in DNTs, practical applications of nano-mass and nano-strain sensors are possible by measuring the DNT frequencies. DNTs can develop wider application in carbon nanomaterials.

## Figures and Tables

**Figure 1 nanomaterials-11-02241-f001:**
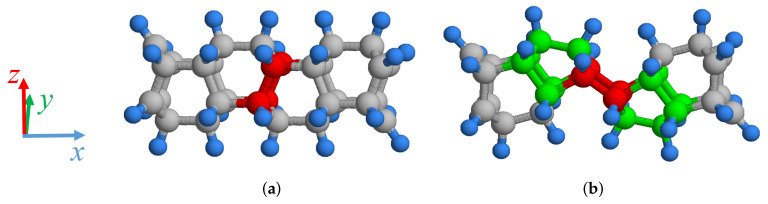
Analysis models of (**a**) perfect DNT and (**b**) DNTs with SW defects.

**Figure 2 nanomaterials-11-02241-f002:**
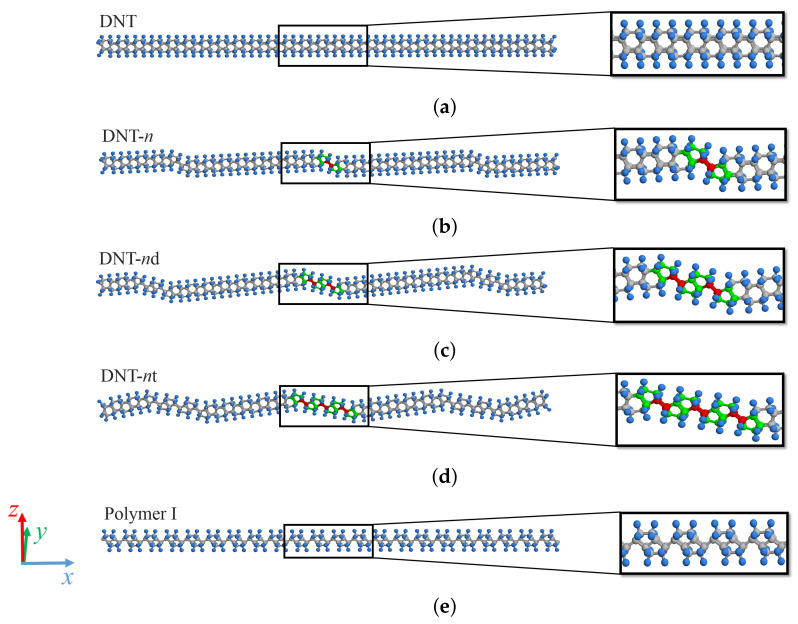
Analysis models of DNTs. (**a**) DNT: perfect DNT. (**b**) DNT-*n*: DNT with isolated SW defects. (**c**) DNT-*n*d: DNT with double SW defects. (**d**) DNT-*n*t: DNT with three SW defects. (**e**) Polymer I.

**Figure 3 nanomaterials-11-02241-f003:**
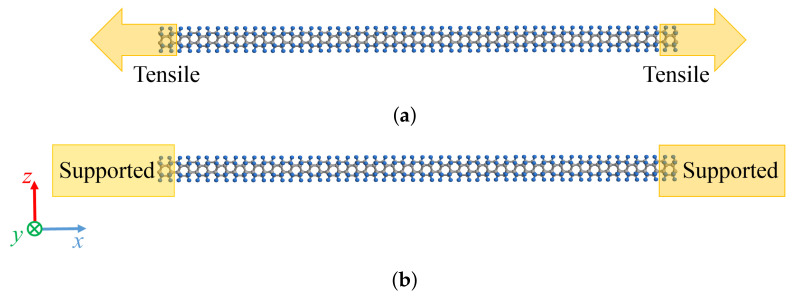
Boundary conditions in (**a**) tensile and (**b**) vibration simulation.

**Figure 4 nanomaterials-11-02241-f004:**
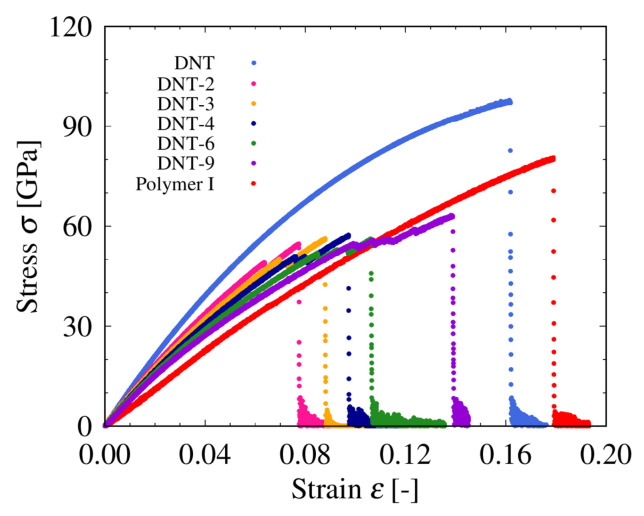
Stress–strain curves of DNTs.

**Figure 5 nanomaterials-11-02241-f005:**
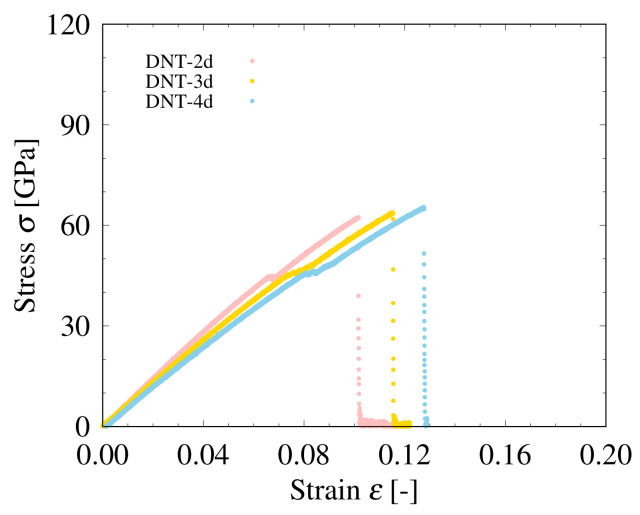
Stress–strain curves of DNTs.

**Figure 6 nanomaterials-11-02241-f006:**
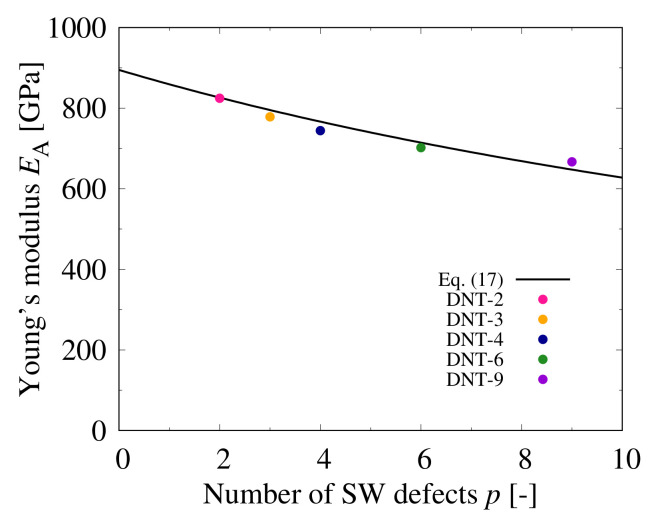
Relationship between Young’s modulus of DNTs and number of SW defects.

**Figure 7 nanomaterials-11-02241-f007:**
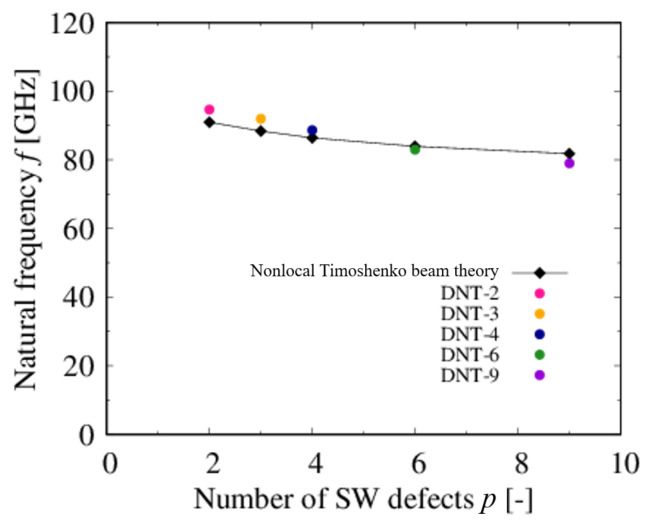
Relationship between natural frequencies of DNTs and number of SW defects.

**Figure 8 nanomaterials-11-02241-f008:**
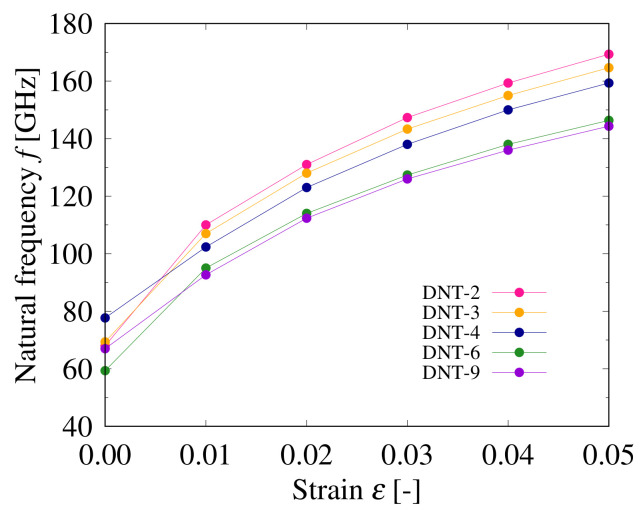
Relationship between natural frequency and applied strain of DNT with isolated SW defects.

**Figure 9 nanomaterials-11-02241-f009:**
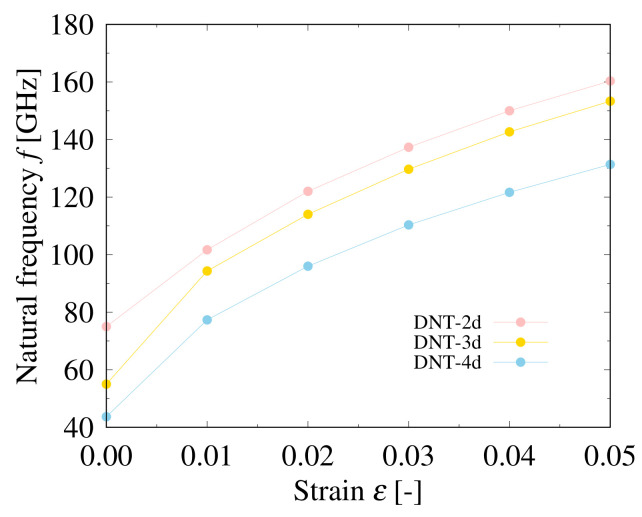
Relationship between natural frequency and applied strain of DNT with double SW defects.

**Figure 10 nanomaterials-11-02241-f010:**
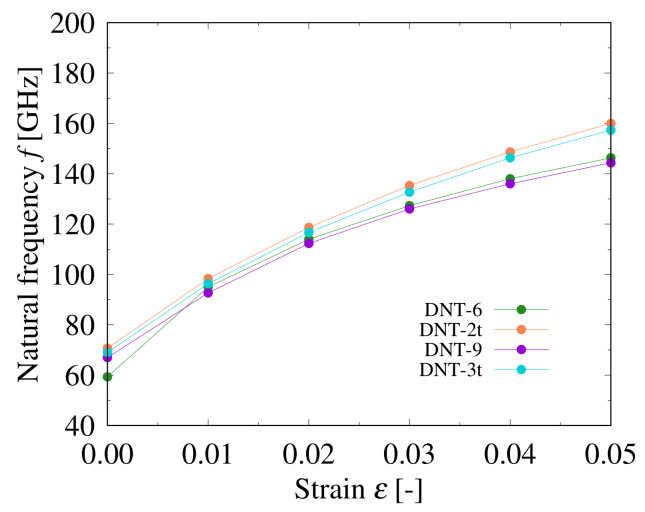
Relationship between resonant frequency and attached mass of DNT with the same number of SW defects.

**Figure 11 nanomaterials-11-02241-f011:**
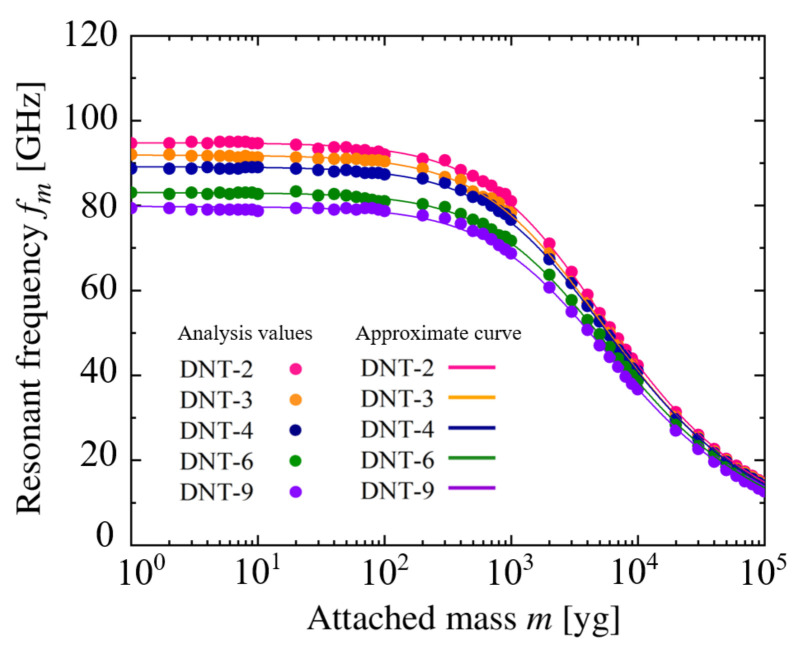
Relationship between resonant frequency and attached mass of DNT with isolated SW defects.

**Figure 12 nanomaterials-11-02241-f012:**
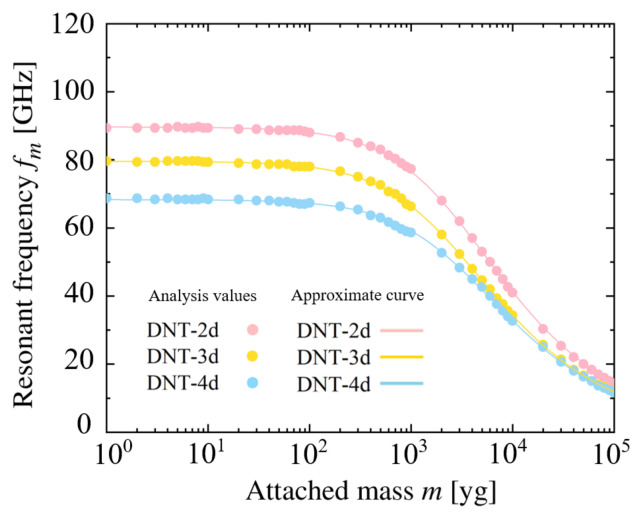
Relationship between resonant frequency and attached mass of DNT with double SW defects.

## Data Availability

Not applicable.

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
