# Peer review of "Vibration Control of Diamond Nanothreads by Lattice Defect Introduction for Application in Nanomechanical Sensors"

_nanomaterials, 2021, doi:10.3390/nano11092241_

Round 1

Reviewer 1 Report

Paper reports a computational study of the vibration properties of diamond nanothreads with SW defects. The study is interesting, but the aspect of novelty is not clear in the text. Results for tensile strain of such nanothreads as function of number/density of SW defects of section 4.1 have been already presented in previous papers. The concept of using nanothreads as nanoresonators is also not new. The only novelty would be the dependence of frequencies with density of defects, which is straightforward to compute according to the model used.

Quite significant revision is needed to become publishable in “Nanomaterials”. Follow below some questions to be answered and suggestions to improve the paper.

1) Having a section named “stone-wales defects” does not make any sense.  Include discussion somewhere else 

2) The name of Section 3 is improper (MD is not an analytical method, it is a numerical simulation technique). More details of MD simulations parameters should be given (timestep, thermostat, barostat)

3) The same theme has been explored previously, as reported in Ref 48 and others. This reference is rarely cited in the manuscript. Authors should compare their results to the ones published by others in this reference, discussing the novelties of the present paper, results that agree with previous ones, etc.. Include a brief discussion in the Introduction and Results sections.

4) Results of tensile analysis should be confronted and compared with previous results of the literature (for example, Refs. 6, 8, 10). Same properties (Young’s modulus, stress-strain curves) for the very same structures have been studied before. How do the results of the authors compare with those from others?  

5) Presentation and discussion of results should be improved, plots in figures seem nice but results deserve a better explanation and discussion. A comparison with a previous study (see remark #3 and 4) and similar properties from other carbon nanomaterials should also be made. What advantages would DNTs bring?

6) Figs 6 and 7 - what does “theoretical values” mean? Was it obtained from fitting the data? explain in the caption. Figs 11 and 12 - correct the typo “apporoximate”

Author Response

We appreciate the encouraging, critical and constructive comments on this manuscript by the reviewer. The comments are very useful for improving the manuscript.  We have taken them fully into account in the revised manuscript and believe that the comments and suggestions should enhance the scientific value of the revised manuscript by many folds. The responses are attached.

Reviewer 2 Report

Molecular dynamics simulation and continuum mechanics approach are here applied to compare mechanical and vibrational properties of “ordered” and defected (Stone-Wales) carbon nanothread (DNT). A good agreement is found between the two methods. The main result concerns with the possibility of changing the resonance frequencies by the introduction of such defects. The authors speculate about the potential use of such defected DNT in practical application such as resonators in nanosensors for detecting for example strain.

The manuscript is interesting and clearly written. The topic is “hot”: a great attention is received by the synthesis of these new exciting materials but there are not yet too many studies about the potential applications. Also for this reason this manuscript deserves attention. My only criticism concerns with the lack of an exhaustive overview of the DNT world. In fact, recently have been synthesized many different threads from furan, thiophene, aniline, pyridine, azobenzene, arene co-crystals and the landscape is now richer and more complex. I think that the entire manuscript would benefit by the addition of this information and would be interesting the author’s speculation about the possible effect of heteroatoms and functional groups (amino or azo groups in the DNTs from aniline and azobenzene respectively) on the properties which are discussed in the manuscript.

Author Response

(The authors gave the same response as above.)

Reviewer 3 Report

The authors focus on investigating the mechanical properties and vibration behavior of Diamond nanothreads (DNTs) without and with lattice defects and examine the influence of density and configuration of using molecular dynamics and continuum mechanics. The paper is overall well written while presents novelty. However the following points should be addressed: There is a confusion about the data presented in the figures. Please identify the MD-only data by using appropriate legends in the figures. Please compare the predicted mode shapes between nonlocal Timoshenko analysis and corresponding trajectories form the MD simulations. How was the additional mass implemented in the MD model? Please justify the use of the specific parameter in the analytical model. For example: why such a nonlocal parameter was selected? I believe that the following works are very relevant with the proposed study and, therefore, will make the state of the art more comprehensive: DOIs: 10.1007/s00707-017-1812-9, 10.1016/j.physe.2013.07.024

Author Response

(The authors gave the same response as above.)

Reviewer 4 Report

The mechanical properties and vibration behaviours of Diamond nanothreads without and with lattice defects are investigated.

The topic falls in the journal scope. The manuscript is organized with diagrams.

Some comments are listed as follows:

  1. Introduction can be improved by adding necessary background information on nonlocal models since in the manuscript a comparison between MD and a continuum nonlocal approach is provided. This section should give some information about non-classical theories such as nonlocal elasticity and stress-driven theories such as On the carbon nanotube mass nanosensor by integral form of nonlocal elasticity, International Journal of Mechanical Sciences, 150, 2019, 445-457; Axial and torsional free vibrations of elastic nano-beams by stress-driven two-phase elasticity (2019) Journal of Applied and Computational Mechanics, 5 (2), pp. 402-413. Variationally consistent dynamics of nonlocal gradient elastic beams (2020) International Journal of Engineering Science, 149, art. no. 103220.
  2. (1) and (2): which is the nonlocal model adopted in these Eqs?
  3. (1) and (2): symbols should be defined.
  4. Figure 7: there is not the Timoshenko beam model.
  5. (6): which is the mechanical meaning of this boundary condition?
  6. Which are the external constraints of the considered nonlocal beam?
  7. Lines 171-174: The nonlocal elasticity of the Eringen model is considered. A comment should be added especially because it is used for the first time to calculate the vibration properties of DNTs.  Accordingly, a nonlocal stress driven model is proposed in the framework of nonlocal elasticity. A discussion can be found in: Random vibrations of stress-driven nonlocal beams with external damping. (2021) Meccanica, 56 (6), pp. 1329-1344 and so on.
  8. 6: the symbol “x” has been adopted for the x-axis. What is the same symbol in the horizontal axis of the plot?
  9. (17): It is not clear how this eq. is obtained by the theoretical analysis.
  10. The novelty of the paper should be better pointed out.
  11. References should be improved in the part on nonlocal models by quoting and commenting additional papers.
  12. Typos should be corrected (such as “stress–train responses”).

Author Response

(The authors gave the same response as above.)

Round 2

Reviewer 3 Report

The manuscript is now appropriate for publication.

Reviewer 4 Report

Accept as it is.